# Macroencapsulation of Paraffin in a Polymer–Gypsum Composite Using Granulation Technique

**DOI:** 10.3390/ma15113783

**Published:** 2022-05-25

**Authors:** Krzysztof Powała, Andrzej Obraniak, Dariusz Heim, Andrzej Mrowiec

**Affiliations:** 1Faculty of Process and Environmental Engineering, Lodz University of Technology, 90-924 Lodz, Poland; andrzej.obraniak@p.lodz.pl (A.O.); dariusz.heim@p.lodz.pl (D.H.); 2Polytechnic Department, Akademia Kaliska, 62-800 Kalisz, Poland; a.mrowiec@pwsz.kalisz.pl

**Keywords:** phase-change material, gypsum, granulation, paraffin, compressive strength, thermal conductivity

## Abstract

This article shows research confirming the thesis on the use of a new material in the form of gypsum, paraffin, and polymer. The article presents an innovative method of preparing plaster with PCM and polymer. Using a special wheel, it was possible to produce a granulate consisting of a mixture of gypsum and paraffin and then spray it with various preparations in order to select the best substance for encapsulation. The article covers strength tests of the obtained granulate depending on the encapsulated material, as well as screening and separation tests depending on the diameter of the granulate. Then, samples consisting of each type of granulate were prepared and poured with gypsum. Studies of the heat conductivity coefficient, the volumetric heat capacity, and thermal diffusivity were carried out. After obtaining the test results, the development of temperature changes was examined for two gypsum boards, one made of raw gypsum and one containing granules, which achieved the best results. The test was carried out using special lamps that were supposed to emit a total of 1000 W of power. The temperature in front of and behind the plates was examined and appropriate conclusions were drawn.

## 1. Introduction

Due to the rapid increase in energy demand, many scientists are trying to find new solutions to improve the energy balance of buildings. Due to the fact that the number of new buildings is constantly growing, it is important to store the energy already produced. As is known, the construction sector is responsible for about 30% of the consumption of energy produced and is still growing, so it is important to develop the technology of energy-efficient buildings. Therefore, it is worth looking at the thermal energy storage technique, which has many advantages. One of them is to design the temperature distribution in such a way that it positively affects the thermal comfort intended for people. With the use of appropriate building materials, a properly selected temperature distribution in the room supports the operation of building materials that not only store thermal energy, which is later used, but also absorb larger amounts of energy when the temperature is increased, e.g., at room temperature in summer, eliminating the influence of air conditioning on indoor temperature, saving electricity.

One of the several ways of storing thermal energy is possibly using phase-change materials [1,2]. It is a known method, developed for over a dozen years, where new solutions and new building materials containing PCM appear. The inclusion of PCM in external partitions ensures that the accumulation of heat in these materials changes the method of heat transfer, which allows for obtaining new parameters of these layers. The work to date shows that the impact of PCM is unequivocal on the reduction of energy consumption and PCM increases the thermal inertia gains in the applied layers [3,4,5]. However, it depends on many parameters of the phase-change material used. The amount of PCM used is of great importance [6], as well as the type of the phase-change material itself. The thermal properties of the material used are also important. These are mainly: thermal conductivity, density, chemical stability, and the temperature of solid–liquid transition [7,8,9,10]. The cost of such material is also important. Because it is quite innovative, the production price is quite high, especially when it comes to microencapsulation.

The method of using latent heat is not the only method of storing thermal energy. There is also the sensible heat method [11]. However, scientists focused mainly on the latent heat method [12]. There are many studies confirming the thesis that the method of storage using latent heat is the best. Moreover, it has found application in existing buildings [13].

Depending on how the thermal energy storage is designed, the appropriate accumulation material should be selected. As already mentioned, phase-change materials are successfully used. There are two main types of phase-change materials: organic and inorganic. Organic materials include paraffins and fatty acids, while inorganic materials include salt hydrates [14]. Each of these materials has advantages and disadvantages that make it possible to determine which of these substances is better suited for application in a particular solution. Thus, organic materials do not have a negative effect in terms of corrosion, there is no supercooling phenomenon, and they are also chemically and thermally stable, which is extremely important in the design of external layers. Unfortunately, in the context of their use in construction, they are highly flammable, such as paraffin, and have a relatively low thermal conductivity coefficient. However, inorganic materials have a high enthalpy of phase change but are not chemically stable and cause corrosion [15]. When looking more closely at individual materials that can be used in construction, paraffin seems to be a good phase-change material. These are alkanes that are characterized by a fairly wide temperature range of the phase change. This allows for the design of the external or internal layer based on the average daily temperature. Paraffins have another advantage: compared to microencapsulation, they are cheap, which allows the use of this material in various proportions in basic building materials [16]. The biggest disadvantage of paraffins is their low coefficient of thermal conductivity, which in the context of thermal energy storage means that the accumulation rate may be unsatisfactory. Then, various additives are used to increase conductivity, e.g., graphite. Salt hydrates are also PCMs that are characterized by a fairly high thermal conductivity coefficient. However, their main disadvantage when used in construction is chemical instability. This can cause chemical degradation at high temperatures. In addition, it causes metal corrosion in reinforced materials, which is unacceptable from construction point of view. Looking at the expectations in terms of the balance of thermal energy and its storage, it is important that the properly selected phase-change material should have several advantages in the context of its use in construction:-Large range of phase change;-Repeatability on phase change;-High specific heat;-High thermal conductivity;-Chemical stability;-Low flammability;-Low cost and high ease of manufacture.

There are also methods of preparing phase-change materials that improve properties. Microencapsulation is a fairly well-known method. The method itself allows, among others, preparing paraffin in such a way that during the phase change, there are no large changes in the volume of the material [17]. In the case of paraffin, there is another disadvantage, quite important in terms of not only aesthetics but also flammability; it is leakage in the liquid state. Microencapsulation also counteracts this disadvantage by sealing paraffin particles in a capsule [18,19]. The idea of microencapsulation is to extract a molecule of the phase-change material and soak it in a tight shell that separates it from the environment. Microencapsulation is performed in several ways. These are physical, physico-chemical, and chemical methods [20]. Spray drying is one of the physical methods [21]. This method is often used to microencapsulate paraffin [21,22]. The most common method for producing microencapsulation is the chemical method. It falls into three categories: interfacial polymerization, suspension polymerization, and emulsion polymerization. 

When comparing the individual microencapsulation methods, each has advantages and disadvantages in manufacturing that affect the performance of the capsules after manufacture. Physical methods, which include spray drying, are cheap and capsules of various dimensions can also be produced. The disadvantages are primarily that the capsules aggregate into larger agglomerates in an uncontrolled manner as well as not all capsules are coated by the process. In the case of mechanical strength, it is low, while the life of the shell itself is unsatisfactory [23]. In the case of physico-chemical methods, coacervation is distinguished, which is characterized by a large control on the size of the capsules produced, but unfortunately the particles aggregate into agglomerates. Similar to spray drying, here the capsules have low strength and the shell’s lifetime is low [24]. Another physico-chemical method is sol–gel microencapsulation. This method has the important advantage that the shell produced has a high coefficient of thermal conductivity. This method is a modern process. Therefore, it is not well developed and additionally generates high costs. However, the mechanical strength of the capsules produced is quite high, while the durability of the shell is satisfactory [25]. The last chemical method that is mainly used for microencapsulation falls into three categories. The first is interfacial polymerization, which is characterized by the versatility of the produced capsules being used for various purposes. Unfortunately, the capsules produced have a low mechanical strength [26]. Another chemical method is suspension polymerization, which has the great advantage of allowing the heat to be controlled during the process. It is also characterized by capsules with good mechanical strength, and the shell has a fairly long shelf life [27]. The third and final chemical method is emulsion polymerization. Similar to capsules produced by suspension polymerization, capsules here have good mechanical properties and the shell of the capsules has a long shelf life. By analyzing all the methods, it can be seen that chemical methods are most often used for microencapsulation. The most important of its advantages is that the capsule has a long shelf life, which is necessary in the construction industry or other industry where PCM is integrated for a long period of time. It is also known that in construction, mechanical strength is important, so in order not to significantly affect strength, chemical methods are a good solution. Each of the above-described methods is quite complicated and in most cases requires the use of specialized equipment, which is often expensive. Therefore, solutions other than microencapsulation are proposed.

Granulation is a widely used method for the production of various types of fertilizers and plant protection products. However, the method itself can be successfully used for many other applications. It is worth remembering that the granulation process is a complex process, especially in an industrial environment. In terms of the laboratory, two interrelated steps can be distinguished. It is a method using a rotating disk and a granulation mixer or only a rotating disk [28]. These methods are closely related to each other. However, attention is drawn to the method with only the rotating disk. This solution is quite simple and cheap to make. The only disadvantage is the higher moisture content of the material and the risk of coalescence of the granules. However, the correct dosage of the binding liquid and control of humidity make the method a good solution for the production of granulated gypsum.

The application of gypsum in construction is mainly plasterboard. There are also different types of gypsum putty, gypsum plaster, or gypsum composites, but each of these forms is mainly used inside the building. Therefore, the gypsum layers are not exposed to weather conditions and various admixtures can be added. Due to the trends in saving thermal energy, one of the solutions is to use a phase-change material for gypsum plasterboard. Research shows [29] that adding PCM is beneficial for several reasons: most importantly, it increases thermal energy storage. Another feature is the reduction of the thermal conductivity coefficient, which positively affects the reduction of thermal energy loss. The research was carried out for several shares of the phase-change material in the plasterboard (5%, 10%, and 15%), which had a negative effect on the compressive strength proportional to the PCM content. Similar considerations were made in the study [30], in which strength tests were also carried out for various shares of the phase-change material. In this case, too, the compressive strength deteriorated due to the addition of more water to grind the PCM depending on the percentage. In conclusion, the reduction in mechanical strength is due to the greater amount of water necessary to mix PCM with the gypsum. Similar conclusions were reached in the study [31], in which the saturation of calcium silicate with water was investigated. The results show the relationship between dry and water-saturated samples in terms of compressive strength. The situation repeats itself because the higher water content reduces the compressive strength. In another research [32], a gypsum–geopolymer mixing test was carried out. It turned out that the polymer influences the setting time of the plaster and the amount of water absorbed. Additionally, a study of the heat conductivity coefficient, the volumetric heat capacity, and thermal diffusivity was carried out. In the case of the thermal conductivity coefficient, the values decreased compared to raw gypsum, while the heat capacity increased. As for the diffusivity, it slightly decreased compared to raw gypsum. Another interesting way to trap too much water when adding PCM is through superabsorbent polymers (SAPs). SAPs are networks of hydrophilic polymers that absorb and hold water [33,34]. This is a good alternative due to the problems with drying the gypsum, which can lead to cracking at high percentages when a phase-change material is added. As you can see, there are many interesting solutions that can be applied to improve water absorption or retain it in the context of deteriorating strength properties of gypsum mixed with PCM. It turns out that polymers work well in this.

The study presented in this article is a continuation of that in [35], which presents a gypsum composite containing paraffin, gypsum, and polymer as a homogeneous mixture. The article presents an innovative method of combining gypsum with paraffin and polymer in the form of granules, which was then subjected to further research. The main points of innovation are primarily the use of wet granulation to produce gypsum granules. The granulation process itself is known, but there are no studies leading to its use in gypsum mortar. In addition, the study prepared a ready-made product consisting of granules and poured with gypsum, thus imitating a plasterboard. First, the pellets themselves were tested for compressive strength. On this basis, it was checked which protective layer was selected from the polymers selected from the previous study, potassium glass, sodium glass, silicate, and water. Then, samples of granules covered with gypsum were prepared as a joint. Thermal conductivity, volumetric heat capacity, and thermal diffusivity were tested for this type of material. The last test confirming the correctness of the granulate used was a temperature test of two materials, raw gypsum and granules with the selected potting material, which achieved the best results in the previous tests.

## 2. Materials and Methods

### 2.1. Tests on Gypsum Granules

Rubitherm RT 22HC paraffin was used in this study. It was chosen because of the temperature phase change point, which oscillates at 22 °C. It was decided to use paraffin with a transformation temperature of 22 °C due to the similar value of the temperature of thermal comfort for people. Technical data are given in Table 1. 

The gypsum used for the tests was a raw material and is characterized by the lack of admixtures, improving, e.g., the hardening speed. The idea was that pure gypsum sieved through 0.5 mm was the basis for adding further ingredients to achieve the desired results. Table 2 presents the most important properties of gypsum.

#### 2.1.1. Production of Granules

The next part of the article describes an innovative method of preparing a gypsum composite. It is an innovative method that was created on the basis of previous research [35] on gypsum composite. To make gypsum granules, the following steps were taken:The appropriate amount of gypsum was poured on a special slow-rotating wheel.Liquid paraffin was then slowly sprayed with a nozzle.The next step was to spray the resulting granulate with one of the 5 substances selected as the protective layer. Spraying was carried out with a spray device with a constant stream of liquid. While the wheel was rotating, the spraying was concentrated in the place of the largest aggregate of gypsum granules. Spraying stopped when coalescence appeared. The obtained granules were clearly more stable, and they were subjected to strength tests in order to select the best reinforcing substance. The following substances were used for this purpose:-A polymer that was previously used for direct addition to form a homogeneous mixture;-Silicate StoPrim—Sto company (Table 3);-Potassium glass—VITROLIQ P—20, Ciech Vitrosilicon company (Table 3);-Sodium glass—VITROLIQ S—130, Ciech Vitrosilicon company (Table 3);-Water as reference.

Separate granules were created for each of these substances. Therefore, 5 series of measurements were prepared. The formed granulate was sieved through sieves ranging from 10 mm to 1 mm (Figure 1). The granulate prepared in this way was allowed to harden for a month.

#### 2.1.2. Compressive Strength Testing of Selected Gypsum Granules

To check the suitability of gypsum granules for use on a real scale, they were subjected to a compressive strength test. An Instron device was used to carry out the strength tests (Figure 2a). The test was carried out in 5 series of measurements, dividing the measurements depending on the material used. As a result of screening the material and separating granules of appropriate diameters, two types of granules were subjected to a strength test. The granules were measured for compressive strength by taking the average of 5 measurements for each type of granulate. The measurement was carried out until the sample was destroyed. Thus, for each material used, polymer, silicate, potassium glass, sodium glass, and water, the compressive strength of the granule was measured at 5 mm and 10 mm, respectively. The selection was made on the basis that granules with a diameter of 10 mm were the largest granules obtained by sieving, while granules with a diameter of 5 mm were selected for the largest proportion in each of the granules produced (Figure 2b).

#### 2.1.3. Studies of Thermal Conductivity, Volumetric Heat Capacity, and Thermal Diffusivity

The tests of thermal conductivity, volumetric heat capacity, and thermal diffusivity were carried out with the use of Isomet 2114. To diversify the measurement of parameters, the samples were conditioned in a special cabinet, allowing for the determination of a specific temperature. Therefore, tests were carried out in the temperature range 10–34 °C. A test was carried out at 1 °C for each type of produced granules.

#### 2.1.4. Testing Temperature Changes in Two Types of Gypsum Boards

The purpose of the test was to find out how the temperature would change in front of the boards from the outside and behind the boards. Gypsum-coated granules were used as they showed the best results. To prepare the plates for measurement, the granulate was poured with gypsum, thanks to which, it allowed to create a uniform surface ready for measurements. The second plate was completely filled with the same raw plaster. The gypsum was mixed with water in a proportion of 0.35:0.65 with a mechanical stirrer. Due to the filling of one board with granulate, 5 kg of gypsum was used, while 10 kg of gypsum was used for the reference board. The boards prepared in this way were dried for 2 weeks. After this time, temperature changes were investigated. In this case, 12 thermocouples were used and more accurate PT 1000 sensors were used to record each change. For more accurate measurements, the results were recorded every 4 s. Two boards 0.5 m × 0.5 m in size and 25 mm thick were prepared (Figure 3a). The dimensions of the boards used for the test imitated the thickness of the gypsum boards for which gypsum granules were mainly intended.

To generate radiation in this study, 8 SYLVANIA lamps were used to generate 1000 W of power (Figure 3b). To check and estimate the distance between the lamps and the tested plates, a pyranometer was used to measure the radiation intensity, thanks to which it was possible to determine the power of 1000 W at a distance of 2.20 m. To collect data, the LabView program was used, which allowed to save measurements simultaneously from 12 sensors. The sensors were directly connected to the NI 9219 controller, where the data then went to the LabView program.

## 3. Results

### 3.1. Tests on Gypsum Granules

#### 3.1.1. Production of Granules

Preparation of gypsum granules is an innovative method of composite preparation and an extension of previous works on it. This method was developed due to the preparation of such granules that can be used in various layers, not only on plasterboard walls. The granules produced in this way are easy to transport and mix with diluted gypsum and retain paraffin in almost 100% of the cases, and as a result of the presented solution of strengthening them with appropriate coatings, they retain satisfactory mechanical strength. Of course, at the beginning of the research, it was necessary to extract the granules of the appropriate diameter. Figure 4 shows that for all five surrounding substances, the main proportion was from 4 mm to 6 mm, which means that the granules are quite fine and easy to apply. This means that the building layer can be designed with different thicknesses without being limited by the granules.

#### 3.1.2. Compressive Strength Testing of Selected Gypsum Granules

Another test was to check whether the produced gypsum granulate meets the requirements of widely used materials. Compressive strength was tested again, and granules similar in shape to a sphere were studied. To verify the granules with each other, the measurements were made of granules with a diameter of 5 mm, which constituted the majority of the proportion, and granules with a diameter of 10 mm, which were the largest granules that could be produced and screened. In Figure 5, it can be seen that pellets with a diameter of 5 mm have a clearly greater compressive strength than pellets with a diameter of 10 mm. In all cases, the strength is twice as high. The polymer turned out to be the best encapsulating material, which means that it was correctly selected for research on gypsum composites. It can be seen that potassium glass is also a good material for protecting granules and also meets the conditions for using such prepared granules on a real scale.

#### 3.1.3. Studies of Thermal Conductivity, Volumetric Heat Capacity, and Thermal Diffusivity

After the compressive strength tests were carried out, further tests were carried out to determine the sense of using gypsum granules. For this purpose, samples were prepared simulating the finished surface, which corresponded to the thickness of a standard plasterboard. Therefore, the granulate was poured with gypsum, thanks to which it created a coherent surface ready for subsequent measurements. It was also a requirement to use a plate measuring probe. The study included tests of the thermal conductivity coefficient, the volumetric heat capacity, and the thermal diffusivity. This made it possible to determine one of the most important thermal parameters on the basis of which the suitability of such materials in construction is determined. This method is interesting because it allowed to check the parameters in a certain temperature range, especially at the melting point of PCM. The method had turned out to be effective in the previous study [35]. Therefore, the same test was carried out for gypsum granules, which will allow the results from previous studies to be compared.

By analyzing Figure 6, it can be seen that each granulate reached its maximum thermal conductivity coefficient in the range of 23–25 °C, around the phase transition temperature of the paraffin used. This means that a phase change takes place during which the thermal conductivity coefficient increases. It can be noticed that the highest coefficient of thermal conductivity the polymer reaches is about 0.82 W/m·K at 23 °C. Compared to other gypsum granules, the difference is considerable, especially when compared to silicate-coated granules, where the value is 0.54 W/m·K at 25 °C.

Since thermal conductivity is related to the volume of heat, the situation is similar in Figure 7. The best results are achieved by polymer-coated granules because it reaches its maximum at 23 °C and amounts to 1.66 MJ/(m^3^·K). It turns out that the granulate coated only with water was slightly worse, reaching its maximum at 24 °C and amounting to 1.59 MJ/m^3^·K. Again, the silicate-coated granulate turned out to be the worst as it remained virtually unchanged over the entire temperature range.

In Figure 8, the best gypsum granule polymer coating effect can be seen again. As in Figure 6 and Figure 7, this granulate reaches its maximum at 23 °C and reaches the value of 0.49 mm/s. The sodium-glass-covered granulate turned out to be slightly worse, reaching the value of 0.48 mm/s at 24 °C. However, analyzing these three parameters, the polymer coating with gypsum granules gives the best results.

#### 3.1.4. Testing Temperature Changes in Two Types of Gypsum Boards

After conducting a series of tests, an attempt was made to choose the best solution that allowed the last test to be performed. Due to the fact that in the strength tests and tests of the thermal conductivity coefficient, the polymer-coated granulate turned out to be the best, it was selected to produce a gypsum board in which the granulate was embedded. For comparison and a better understanding of the sense of using PCM, the test was diversified by adding a panel with raw gypsum. Thanks to 8 lamps, solar radiation with a power of 1000 W was simulated, thanks to which 2 plates were subjected.

By analyzing Figure 9, it can be seen that the board containing the PCM granulate and the polymer coating embedded in the plaster has better thermal properties. First of all, it can be noticed that in 3 H of heating, there is a phase change around 22 °C. It can also be noticed that there is a time shift at the moment of reaching a given temperature. It is best seen at the temperature of 25 °C, where the shift from reaching this temperature is 1 H in relation to the raw gypsum board. The peak temperature is also reduced to approx. 3 °C. However, in the second phase of the experiment, in which the lamps were turned off and allowed to cool down, it can be noticed that the temperature decreased more slowly for the PCM plate, which positively affects the entire cycle of heating, storage, and heat dissipation inside building. Analyzing the results determining the temperature outside the surface of the plates, it can be seen that the PCM plate also heats up slower, but it is not such a noticeable difference. The situation is similar for plate cooling, where the temperature decreases slower for a PCM plate, but the difference is also not large. Nevertheless, the study shows that the granulate used increases the thermal properties of the gypsum board.

## 4. Discussion

This article is an extension of the work on a gypsum composite, initially in a homogeneous form, and in this case as gypsum granules. When comparing the two methods, it can be seen that each of these methods is quite simple and cheap to implement. The gypsum composite, being a mixture of paraffin, gypsum, and polymer, had one major advantage. Raw gypsum is quite porous, which causes high water absorption, which adversely affects the surface. This state of affairs is influenced by the degree of porosity of the material. In the conducted studies [36], the apparent porosity is reduced by adding non-encapsulated PCM. The direct addition of paraffin has been found to reduce the porosity, explained by the fact that the paraffin fills the pores in the material without leaving many voids. This behavior may be due to the fact that the paraffin acts as a hydrophobic material and the porosity is lower than that of PCM-free gypsum. The same study also investigated the effect of microencapsulated PCM, where it was found that paraffin capsules increase porosity, which is disadvantageous. However, the surface prepared in this way is characterized by better thermal properties, such as a higher heat transfer coefficient.

Comparing the individual test results of other scientists with the results obtained in the compressive strength tests, it can be observed that the results are satisfactory. The concept of using a polymer as a sealing and reinforcing material gives good results. In the previous tests [35], tests of annealing and cooling of the material were carried out. They gave good results because the weight reduction was at the level of 5%, and as you know, paraffin changes its aggregate state at a certain temperature. However, it was not possible to avoid paraffin leakage, which was also attempted in another study [37]. Moreover, there were problems with the mixing of plaster and paraffin in the study. The test was also extended by the compression and bending strength test of plaster not only with paraffin but also with salt hydrates. Unfortunately, salt hydrates showed a high loss of compressive strength, of 49.9%, and bending strength of 55.5%. For comparison, gypsum with paraffin had a reduction in compression of 4.2% and a bending reduction of 9.4%. Another interesting study [38] was the investigation of gypsum with various admixtures. The following elements were added to the gypsum: microspheres, which are products of coal combustion; aerogel, which is characterized by a low thermal conductivity coefficient; and a polymer characterized by water resistance, widely used in construction. The results of the compressive and bending strength tests confirm the thesis that also in this case the samples with admixtures were weaker than with raw gypsum. However, it was noted that the deterioration was mainly due to the variable density and porosity. As established in the study, the reduced compressive strength and flexural strength were due to the reduced density of the composite. The amount of phase-change material is also of great importance for the reduction of the compressive strength. In the study [39], 10%, 20%, and 30% share of PCM were taken into account. Although the research was carried out for various mixtures of gypsum with fibers and phase-change material in the form of microencapsulation, the tendency was maintained on individual days. Gypsum samples with 20%, 30%, and 40% share were tested for compressive and bending strength after 28 days and 56 days. The trend was maintained in each study and was equal depending on the amount of PCM. This means that it is not the amount of phase-change material that causes a significant reduction in strength but the lack of other admixtures in this case of fibers.

Comparing the direct method of mixing plaster with paraffin and polymer to the method with microencapsulation, a significant advantage can be noted: ease of implementation and cheap solution. However, this solution gives lower thermal properties and there is also a problem with paraffin leakage. Therefore, an innovative method of gypsum granulation was presented that combines both methods. First of all, granules are made by directly combining gypsum and paraffin and then enclosing them in a polymer shell, which is considered the best material. Thanks to such granules, it is possible to use a large amount of paraffin as a binder and increase the thermal properties and create a coating that, as in the case of microencapsulation, prevents paraffin leakage during phase change. Such granules can be easily incorporated in large quantities into gypsum mortar and then used in plasterboards. Therefore, two tests simulating the surface of such a plate were prepared. After introducing the granules into the appropriate molds for specific tests, they were poured with gypsum as a binder. Thanks to this form, tests were carried out to determine thermal properties such as the thermal conductivity coefficient and temperature changes during plaster exposure to the so-called artificial sun with a power of 1000 W.

## 5. Conclusions

This article presents a completely new method of combining plaster with PCM, additionally reinforced with polymer. The method is somewhat related to microencapsulation but does not require the use of specialized equipment and is simple and cheap to perform. Gypsum granules were prepared, which were initially encapsulated with several additional substances. On this basis, the following conclusions were drawn:The granulate, which was prepared on the basis of gypsum and paraffin, was coated with five different substances (polymer, silicate, sodium glass, potassium glass, and water in order to check the effect of hydration with water. After the granulate was formed, it was sieved to obtain granules of various diameters. Granules with a diameter of 5 mm and 10 mm were subjected to strength tests. On this basis, it was determined that in both cases, the polymer was the best reinforcing substance.Thermal conductivity, volumetric heat capacity, and diffusivity were also tested. This time, to test not only the granules themselves, samples were prepared referring to the finished product, which was the gypsum surface. The granules encapsulated with each of the above-mentioned substances were poured with gypsum, thanks to which, after they were taken out of the molds, a homogeneous surface was created. Thermal tests were carried out thanks to the final form of a gypsum surface with granules inside. The polymer turned out to be the best encapsulating material because it achieved the highest thermal conductivity at the transition point. This means that the presented surface containing polymer-coated granules can store thermal energy at the fastest rate.The final test of the presented method of combining plaster with paraffin was the test of temperature changes over time. Therefore, based on previous studies, it was considered that the granulate with the best results would be taken into account. A stand was prepared to imitate sunlight. For a better understanding of the operation of PCM with gypsum, the results were compared to a gypsum board without any admixtures. Based on the results, it can be determined that the slab with granules passed heat more slowly, which means that it heats up slower from the inside, which in application reduces the maximum temperature during the day. When the lamps were turned off, the plate containing the granules decreased its temperature more slowly, which means that the heat was stored and will be given up over a longer period. This undoubtedly means additional thermal benefits at night, which will also have a positive impact in that it will lower the use of the heating system.

By analyzing the presented solution and the research results, it was determined that gypsum granules can be a good alternative to the use of PCM in gypsum. It is important from the point of view of application in plasterboard, i.e., in internal layers between which people live. First of all, thanks to the granulate, the biggest problem, which was paraffin leakage, was eliminated. In addition, water is not used in the process of creating granules, so further savings are made. Nevertheless, the greatest advantage is the cost of producing such a granulate, many times lower than the cheapest method of microencapsulation, which in today’s times of continuous growth of materials and technologies may turn out to be extremely attractive.

## Figures and Tables

**Figure 1 materials-15-03783-f001:**
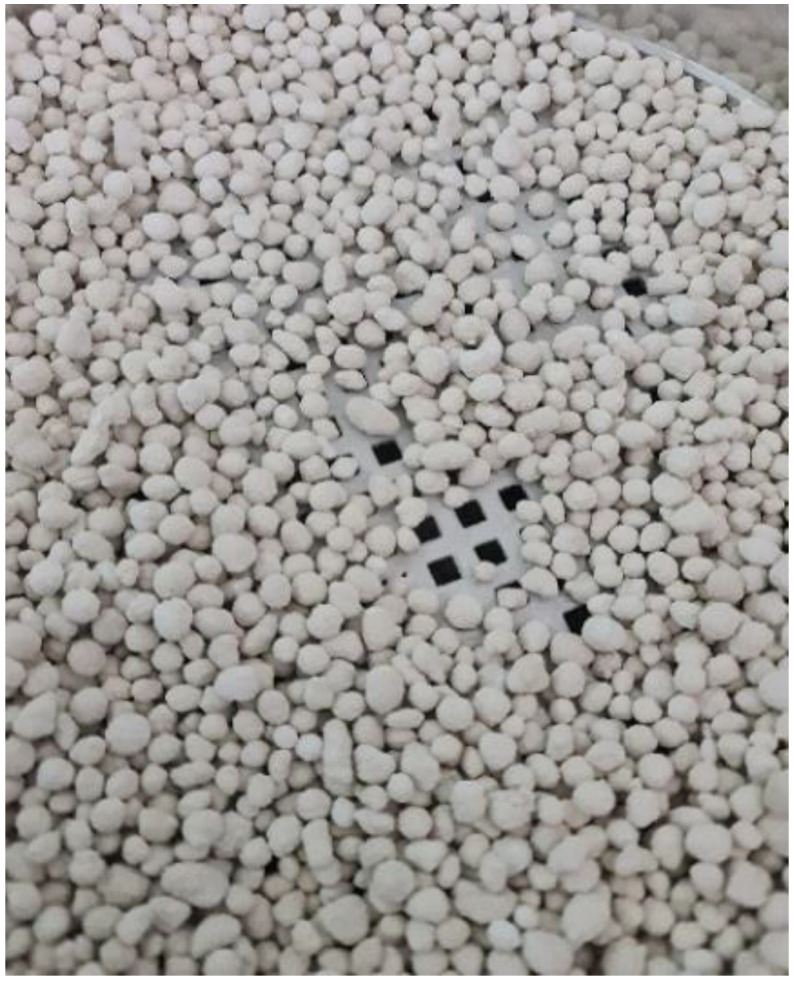
Gypsum granules during sieving through successive sieves.

**Figure 2 materials-15-03783-f002:**
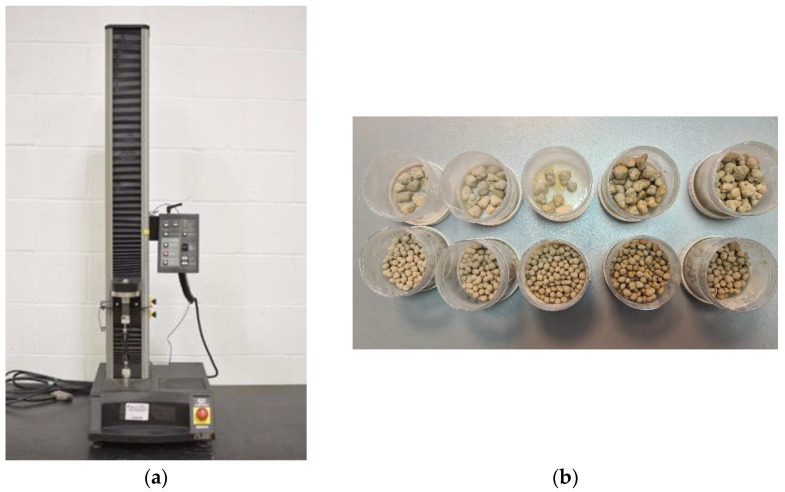
(**a**) Instron device used for compressive strength measurements. (**b**) Granulate samples with a diameter of 5 mm and 10 mm intended for compressive strength tests.

**Figure 3 materials-15-03783-f003:**
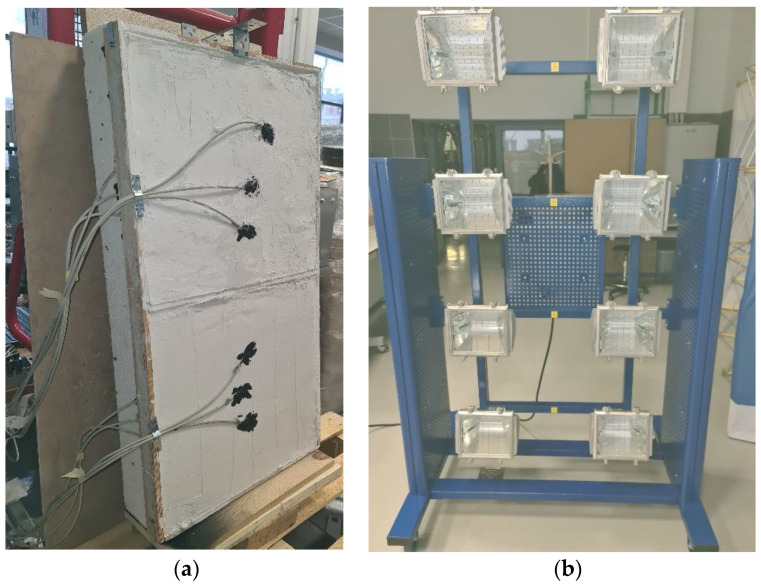
(**a**) Gypsum boards attached to the measuring station; (**b**) SYLVANIA lamps generating 1000 W of power.

**Figure 4 materials-15-03783-f004:**
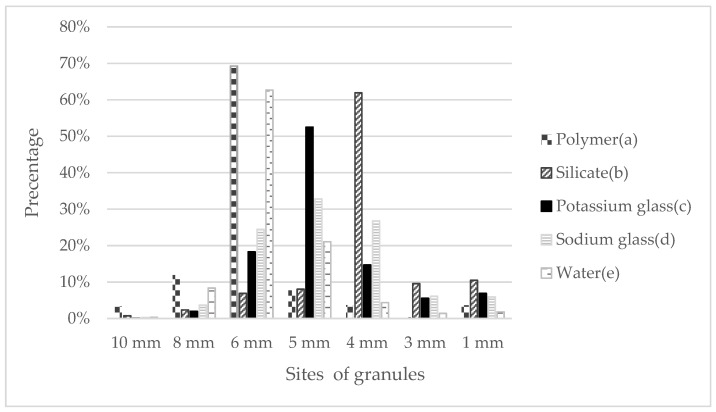
The sifted granules, depending on the diameter, coated with: (a) sodium glass; (b) potassium glass; (c) silicate; (d) polymer; and (e) water.

**Figure 5 materials-15-03783-f005:**
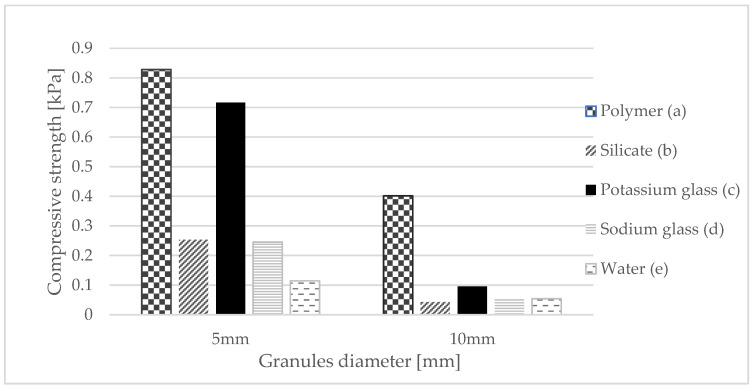
Compressive strength of granules 5 mm and 10 mm in diameter, coated with (a) polymer; (b) silicate; (c) potassium glass; (d) sodium glass; and (e) water.

**Figure 6 materials-15-03783-f006:**
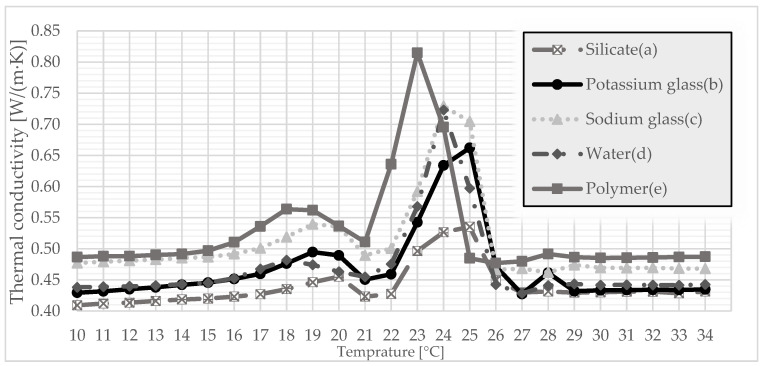
Coefficient of thermal conductivity for a granulate coated with: (a) silicate; (b) potassium glass; (c) sodium glass; (d) water; and (e) a polymer.

**Figure 7 materials-15-03783-f007:**
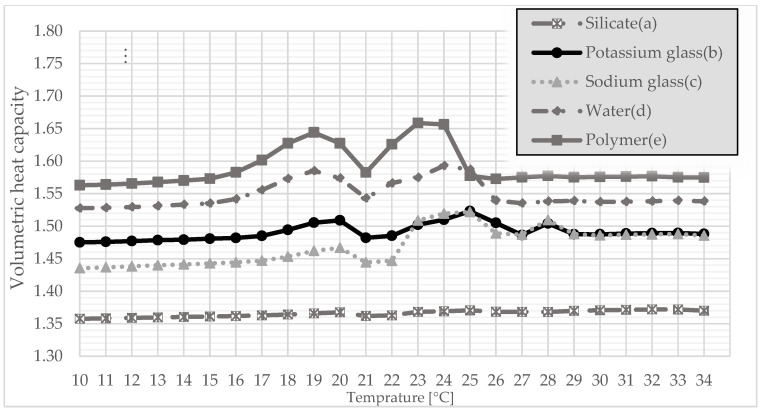
Volumetric heat capacity for coated granules: (a) silicate; (b) potassium glass; (c) soda glass; (d) water; and (e) polymer as a function of temperature.

**Figure 8 materials-15-03783-f008:**
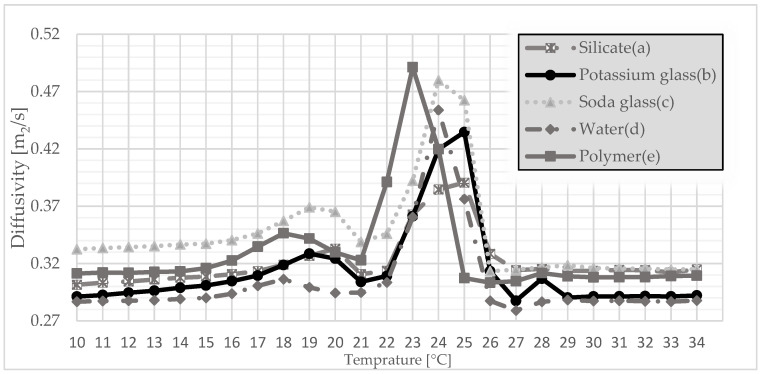
Thermal diffusivity of coated granulate: (a) silicate; (b) potassium glass; (c) soda glass; (d) water; and (e) polymer as a function of temperature.

**Figure 9 materials-15-03783-f009:**
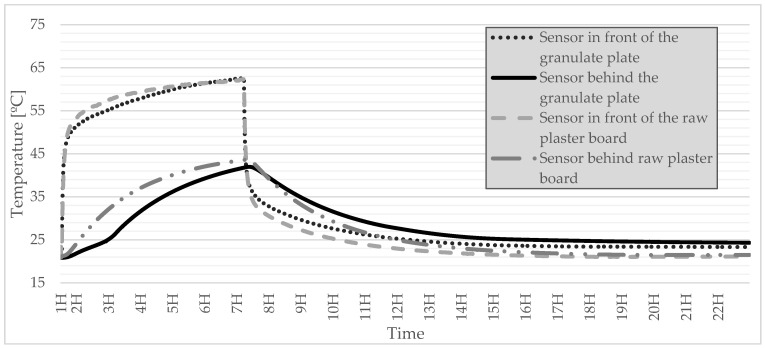
Change of temperature as a function of time for a PCM gypsum board and a raw gypsum board.

**Table 1 materials-15-03783-t001:** Main parameters of Rubitherm RT 22HC paraffin.

Parameter	Unit	Result
Melting area	°C	20–23
Congealing area	°C	23–20
Heat storage capacity	kJ/kg	190
Specific heat capacity	kJ/kg·K	2
Density solid	kg/L	0.76
Density liquid	kg/L	0.7

**Table 2 materials-15-03783-t002:** Main parameters of Atlas gypsum.

Parameter	Unit	Result
Calcium sulfate hemihydrate content	%	>95
(β-CaSO_4_·0.5H_2_O) Crystallization water	%	5.6–6.0
Mechanical strength after drying to constant weight - for bending - for compression	MPa	>5.0 >12.0

**Table 3 materials-15-03783-t003:** The most important data of the substances used in the production of gypsum granules.

Parameter	Unit	Silicate StoPrim (Sto Company)	Potassium Glass—VITROLIQ P—20, Ciech Vitrosilicon Company	Sodium Glass—VITROLIQ S—130, Ciech Vitrosilicon Company
Molar modulus	-	No data	4	3.45–3.60
Weight modulus	-	No data	2.56	3.34–3.48
Density at 20 °C	g/cm^3^	1.05	1.15–1.20	1.335–1.345
Dynamic viscosity at 20 °C	mPa·s	190	3–8	50–70
pH at 20 °C	-	11.0–11.5	No data	No data

## Data Availability

The data presented in this study are available from the corresponding author upon reasonable request.

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
