# Peer review of "Macroencapsulation of Paraffin in a Polymer–Gypsum Composite Using Granulation Technique"

_materials, 2022, doi:10.3390/ma15113783_

Round 1
Reviewer 1 Report
Interesting article integrating PCMs in plasterboard I have a few comments. line 206, you are talking about a new method, have you used a standard to see help? line 211, you are talking about spraying, can you detail the process? line 207, you are referring to earlier references, which ones? line 234, how did you measure the compressive strength? do you have any pics of the device? standard? can you detail? line 249, what type of temperature sensor did you use? thermocouple? Paragraph 2.1.4, information is missing on the formulation of the mixture (quantity of materials, water) How did you mix, implement? Is there a drying time to wait before testing? or a cure? Why these panel sizes? Be careful to check the formatting of the bibliography.Author Response
First, I would like to thank you for the review and the time you spend writing it. I'm glad my
job is interesting.
line 206, you are talking about a new method, have you used a standard to see help?
I did not use the standards to perform this method, it is my original method that allows the
production of gypsum-based granules.
line 211, you are talking about spraying, can you detail the process?
Spraying with paraffin followed by polymer was by a steady stream sprayer. On the other
hand, spraying was inhibited when there was coalescence between the granules.
line 207, you are referring to earlier references, which ones?
This is my mistake, thank you for pointing out. I meant my previous article related to this
article :
Powała K, Obraniak A and Heim D 2021 Testing a Gypsum Composite Based on Raw
Gypsum with a Direct Admixture of Paraffin and Polymer to Improve Thermal Properties
Materials 14 3241
line 234, how did you measure the compressive strength? do you have any pics of the
device? standard? can you detail?
The compressive strength test was performed using the Instron device, which is intended for
testing such samples. The test looked like this, so that the measurement was reliable, 5 tests
were performed for each type of granulate and size of granulate. Thanks to this, the average
was calculated. I was not guided by standards in this case.
line 249, what type of temperature sensor did you use? thermocouple?
Yes. Thermocouples were used in the test.
Paragraph 2.1.4, information is missing on the formulation of the mixture (quantity of
materials, water) How did you mix, implement? Is there a drying time to wait before
testing? or a cure? Why these panel sizes?
This is also my mistake. The boards were made in such a way that raw gypsum was poured
over the granules in the granulate board. 5 kg of gypsum were used for this purpose. The
gypsum was mixed with a water ratio of 0.35: 0.65. The second plate was made on the basis
of 15 kg of plaster with the same proportion. The gypsum was mixed with a mechanical
agitator. The boards took 2 weeks to dry. The dimensions of these boards, and especially the
thickness, were to imitate a plasterboard.
Be careful to check the formatting of the bibliography.
Thank you for your attention, but I don't understand. The bibliography was implemented with
Mendeley.
All changes are highlighted in yellow.

Reviewer 2 Report
The objective of this paper is interesting for the world of materials, providing a new encapsulation way for paraffin. However, the authors should improve some aspects of it:
Section 1. Introduction: Authors have made a brief presentation of the state of the art, incorporating a sufficient number of scientific papers, but without commenting on the conclusions reached. As a recommendation, they could add more specifically the differences or novel points that their work explores with respect to previous ones.
Section 2. Authors have developed and explained in a coherent and systematic way the processes that have been carried out in the study.
Section 3. The authors are recommended to redesign Figure 4 to achieve greater homogeneity with the rest of the document. As well as the legends of Figures 6, 7 and 8. The discussion could be incorporated to point 3, however, this is a minor change, it is not necessary.
Section 4. Conclusions seem to me to be adequate with a brief initial description and the key points that have been reached and that can be expanded in their corresponding section.
Author Response
At the beginning, thank you for your review and I am glad that my work and results are
interesting.
Section 1. Introduction: Authors have made a brief presentation of the state of the art,
incorporating a sufficient number of scientific papers, but without commenting on the
conclusions reached. As a recommendation, they could add more specifically the differences or
novel points that their work explores with respect to previous ones.
1. Thank you for summarizing chapter 1. The innovation in this work consists in the use
of wet granulation in the context of gypsum mortar and the preparation of such
granulate. The process itself is well known, but there are no publications that say that
granulation can be carried out by spraying paraffin and polymer.
Section 2. Authors have developed and explained in a coherent and systematic way the
processes that have been carried out in the study.
2. Thank you for your opinion on chapter 2. I'm glad it's legible and clear.
Section 3. The authors are recommended to redesign Figure 4 to achieve greater homogeneity
with the rest of the document. As well as the legends of Figures 6, 7 and 8. The discussion could
be incorporated to point 3, however, this is a minor change, it is not necessary.
3. Thank you for your opinion on chapter 3. I redesigned chart 4 with your
recommendations, but I did not change the legends for the other charts, they were
unified in terms of layer and color. As for combining the chapter with the discussion, I
think it is better to keep the integral breakdown of the article into this section.
Section 4. Conclusions seem to me to be adequate with a brief initial description and the key
points that have been reached and that can be expanded in their corresponding section.
4. Thank you for your opinion in chapter 4. I am glad that the conclusions are clear.
All changes are highlighted in yellow
Round 2
Reviewer 1 Report
I accept the corrections made. Sincerely